# Improving Upper Limb and Gait Rehabilitation Outcomes in Post-Stroke Patients: A Scoping Review on the Additional Effects of Non-Invasive Brain Stimulation When Combined with Robot-Aided Rehabilitation

**DOI:** 10.3390/brainsci12111511

**Published:** 2022-11-07

**Authors:** Antonino Naro, Rocco Salvatore Calabrò

**Affiliations:** 1Stroke Unit, AOU Policlinico G. Martino, 98122 Messina, Italy; 2Neurorehabilitation Unit, IRCCS Centro Neurolesi Bonino Pulejo, 98123 Messina, Italy

**Keywords:** stroke, robotic, non-invasive brain stimulation

## Abstract

Robot-aided rehabilitation (RAR) and non-invasive brain stimulation (NIBS) are the two main interventions for post-stroke rehabilitation. The efficacy of both approaches in combination has not been well established yet. The importance of coupling these interventions, which both enhance brain plasticity to promote recovery, lies in augmenting the rehabilitation potential to constrain the limitation in daily living activities and the quality of life following stroke. This review aimed to evaluate the evidence of NIBS coupled with RAR in improving rehabilitation outcomes of upper limb and gait motor impairment in adult individuals with stroke. We included 18 clinical trials in this review. All studies were highly heterogeneous concerning the technical characteristics of robotic devices and NIBS protocols. However, the studies reported a global improvement in body structure and function and activity limitation for the upper limb, which were non-significant between the active and control groups. Concerning gait training protocols, the active group outperformed the control group in improving walking capacity and recovery. According to this review, NIBS and RAR in combination are promising but not yet largely recommendable as a systematic approach for stroke rehabilitation as there is not enough data about this. Therefore, more homogenous clinical trials are required, pointing out the best characteristics of the combined therapeutic protocols.

## 1. Introduction

Multiple strategies have been developed to enhance the post-stroke spontaneous recovery mechanisms. These include early reperfusion therapies (i.e., intravenous thrombolysis and mechanical thrombectomy) aimed at limiting damage and preventing further cell death to contain lesion size and disability [1]. Furthermore, traditional (neurofacilitation or functional retraining through either shaping or task practice) and advanced rehabilitation protocols, including pharmacological manipulation to increase sprouting and anatomical plasticity, non-invasive brain stimulation (NIBS) to modulate the activity of targeted brain areas, and robot-aided rehabilitation (RAR) to perform an intensive, repetitive, assisted-as-needed, and task-oriented motor practice, are available in any phase of the post-stroke recovery process [2,3,4,5,6,7,8,9]. These rehabilitation strategies aim to increase the adaptive plasticity processes (mainly experience-dependent plasticity mechanisms) that develop in lesional and perilesional tissues [10,11,12]. 

To date, NIBS and RAR represent two cornerstones of the modern post-stroke rehabilitation era. Both strategies have been employed singularly concerning post-stroke rehabilitation with valuable positive results [2,3,4,5,6,7,8,9]. Both strategies aim at potentiating neuroplasticity mechanisms supporting functional recovery via bottom-up (RAR) and top-down (NIBS) mechanisms [13,14,15]. Bottom-up approaches mainly act at the physical level and attempt to bring about changes at the level of the central neural system, whereas top-down approaches (comprising serious exergames, virtual reality, robots, brain–computer interfaces, rhythmic music, and biofeedback) attempt to stimulate the brain more directly to elicit plasticity-mediated motor relearning [14]. Therefore, RAR and NIBS act indirectly and directly, respectively, on the spontaneous recovery mechanisms occurring after a brain injury (including stroke), which are aimed at substituting a part of the brain for the function of another (according to the theory of vicariation) [10]. In particular, directly modifying the spontaneous recovery mechanisms is postulated to: (i) favor remote structures’ reconnection to the site of injury following the diaschisis period (i.e., a temporary period of depressed metabolism and blood flow), including the perilesional cortex, spared areas in the injured hemisphere, and contralateral homologous and non-homologous areas; (ii) favor the learning of new, compensatory joint and muscle kinematic patterns; and (iii) avoid a maladaptive plasticity process potentially occurring during spontaneous local and sometimes distant rewiring of neural networks (through long-term potentiation, long-term depression, unmasking, synaptogenesis, dendritogenesis, and functional map plasticity) [10,16,17]. These effects occur through targeting the lesioned circuit to foster adaptive connections and minimize faulty connections by providing sensorimotor inputs to the lesioned network designed to specifically foster connections in keeping with Hebbian learning mechanisms (bottom-up approaches). 

RAR, including exoskeletons and end-effector devices, may boost neural plasticity and functional recovery by providing patients with intensive, repetitive, assisted-as-needed, and task-oriented motor practice, which achieves functional motor relearning through the repetitive practice of all different phases of gait and movements of upper limbs related to functional tasks. This effect on neural plasticity is in common with conventional physiotherapy approaches. Actually, training the same movement repetitively enables the nervous system to develop circuits for better communication between the motor center and sensory pathways, which promotes motor function recovery [15,18]. Treatment by RAR compared with conventional treatment presents several advantages, including training duration, more reproducible symmetrical gait patterns, operation by a single therapist, and a reduction in the energy expenditure imposed upon the therapists [18,19,20].In particular, RAR produces benefits similar, but not significantly superior, to those from usual care for improving upper limb functioning and disability in patients diagnosed with stroke within six months. Conversely, recent research revealed that RAR results in a more symmetrical muscle activity pattern in paretic patients compared with conventional treatment, an improvement in activities of daily living, and in lower limb functions and muscle strength, and gait performance [15,18,19,20].

NIBS, including transcranial direct current stimulation (tDCS) and transcranial magnetic stimulation (TMS), can potentiate the neuroplasticity mechanisms entrained by rehabilitative training through associative plasticity mechanisms. At the same time, NIBS can serve as a primer to make the neuroplasticity mechanisms ready to be boosted by rehabilitative training. 

The usefulness of conjugating both approaches stems from the idea that targeting the plasticity mechanisms mentioned above in a very specific manner, i.e., by providing cortical stimuli (top-down approaches) that focus on the effects of bottom-up approaches, could result in a significant enhancement of the plasticity-dependent recovery mechanisms and, eventually, motor function recovery. Some trials aimed to demonstrate the usefulness of these combined approaches concerning both upper and lower limb motor function recovery [3,9,14,21,22,23,24]. However, conclusive data on the efficacy of NIBS and RAR in combination is still missing. Therefore, we aimed to review the trials for upper and lower limb motor function recovery comparing the additional effects of NIBS when combined with RAR vs. stand-alone RAR concerning gait and upper limb function improvement in patients with stroke.

## 2. Materials and Methods

The present review was performed according to the Preferred Reporting Items for Scoping Reviews (PRISMA-ScR) guideline (Appendix A). The literature research was carried out on PubMed/MEDLINE (Medical Literature Analysis and Retrieval System Online; through the PubMed interface), Cochrane Central Register of Controlled Trials (CENTRAL), Scopus, Web of Science (WOS), and PEDro (Physiotherapy Evidence Database) scientific databases, using ‘stroke’ AND ‘robotic OR robotic therapy’ AND ‘non-invasive brain stimulation OR TMS OR tDCS’ as keywords. No publication date or language restrictions were imposed. 

A first search using the abovementioned keywords returned 6281 papers; 2692 articles were removed before screening (Figure 1). The retrieved studies (n = 3589) were screened according to the inclusion/exclusion criteria as follows: (i) randomized, double-blind, sham-controlled (for NIBS), intention-to-treat analysis, clinical trials in adult humans with post-stroke upper/lower limb paresis; (ii) improvement in upper limb motor function and gait as the main outcome; and (iii) use of RAR and NIBS in combination. In addition, papers focusing on infratentorial stroke, applying NIBS other than tDCS and TMS, and not English-written, were excluded from review inclusion.

The so-screened studies (n = 897) were further reviewed on the title and abstract, and 58 articles were identified for full-text reading. After such an assessment, 18 publications were included in this scoping review, 11 regarding upper limb rehabilitation, and seven on gait rehabilitation.

## 3. Results

The main characteristics of the included studies are summarized in Table 1 and Table 2 for the upper [25,26,27,28,29,30,31,32,33,34,35] and lower limbs [36,37,38,39,40,41,42], respectively. The studies were published between 2011 and 2020. The majority of the included studies were randomized clinical trials (RCT) with parallel or crossover design; some among them were randomized studies limited to stimulation sites (i.e., cerebellum or spinal cord) and timing (before, during, or after RAR), and stimulation type (anodal and cathodal) [33,37,38,39]; there were also some differences in the double-blind design concerning concealment allocation, blinding of outcome assessment, and blinding of participants and personnel. 

All studies concerned adults (ranging from eight to 41 individuals, 60–70 years aged on average) with subacute (less than six months) [29,31,32,35,42] or chronic (from 10 to 152 months) [25,26,27,30,33,35,36,37,38,39,40,41] post-stroke disability (regardless of severity, residual motor function, time since last stroke, type of stroke, or history of previous strokes), with an approximately equal distribution of cortical, subcortical, and mixed lesion localizations, and a smaller proportion of lacunar strokes (about 10%). The studies, however, differed concerning sample homogeneity (similarity between characteristics data of active and control groups), representativeness (absence or presence of exclusion criteria other than those usually present in RAR or NIBS trials), and description of sample calculation. Cerebellar strokes or strokes in cerebellar pathways were not included. Both in-patient and out-patient rehabilitation settings were considered. 

Patients were provided with RAR associated with NIBS, including a control group with characteristics comparable to the experimental group, which was provided with RAR paired with sham NIBS. This design was consistent with the purpose of the included studies of assessing whether NIBS and RAR was superior to RAR alone. However, one study [37] also included a third control group (no NIBS), whereas some other studies also included a comparative analysis between different types [35,39,40,42] and timing of delivery of NIBS [33,38]. All included studies also provided the subjects with conventional therapy (including muscle strengthening, joint mobilization exercises, and a comprehensive physical rehabilitation program). 

RAR and NIBS widely varied among studies. Exoskeletons [32,35,36,37,38,41,42] and end-effectors [25,26,27,28,29,30,31,33,35,39,40] were used for RAR, targeting one or more upper limb joints for both unimanual and bimanual training, with different types and degrees of arm-weight support (Table 1), and both lower limbs for gait training, with different degree of body-weight support (Table 2). tDCS was adopted for NIBS before [25,28,33,36,38], during [26,29,30,31,32,33,34,35,37,39,40,41,42], or after [33,38] the RAR sessions; usually, patients were provided with 7 to 36 sessions. One study reported on the continuous theta-burst stimulation effects [27]. tDCS setup varied among studies for electrode dimension (12.56 cm^2^ for cerebellar tDCS, 23.75 cm^2^ for spinal tDCS, and 25 vs. 35 cm^2^ for tDCS over M1), position (the cathodal tDCS mainly being used over the unaffected and the anodal tDCS or continuous TBS over the affected cortical motor areas corresponding to upper or lower limb); some studies implemented both anodal and cathodal tDCS (namely, bilateral tDCS) [25,26,31,38]), and shape (circular vs. rectangular). The stimulation intensity varied from 1.5–2 mA for tDCS over M1 to 2 mA for cerebellar tDCS and 2.5 mA for spinal tDCS; the duration ranged from 7 to 20 min (Table 1 and Table 2). 

Subjects were generally evaluated for body function and structure using the Fugl-Meyer Assessment, whereas Box and Blocks Test, Wolf Motor Function Test, and Action Research Arm Test were used concerning activity limitation. Only a few studies included neurophysiological outcomes [33,38]. Nearly half of the studies assessed the spatiotemporal gait parameters. Follow-up measurements were taken at post-intervention (all studies), two weeks [39], one month (all lower limb studies), and three months [38]. The studies differed concerning description or implicit intention-to-treat analysis, the extent of loss, information regarding early cessation of trials, and selective reporting. The reviewed works were not supported by any grant from the public or private sector, there was nothing to disclose financially, or information funding was unavailable, but for one study [41], which the robot manufacturer sponsored. 

Overall, the studies reported that NIBS with RAR intervention was not superior to stand-alone RAR (with some exceptions that documented the superiority of the combined approach) concerning the improvement in upper limb body structure and function (Table 1 and Table 2) [3,9]. Conversely, the combined approach outperformed stand-alone RAR with regard to gait recovery; however, some specific differences were appreciable between groups when employing spinal and cerebellar tDCS [39,40] and when comparing bilateral tDCS performed before, during, and after RAR [33,38].

## 4. Discussion

Restoring motor function is still a challenging issue in post-stroke rehabilitation. Most stroke survivors present a reduced ability to walk and limited upper limb activities in indoor and outdoor settings, with a poor quality of life. Although gait recovery is usually more easily achieved compared to upper limb function recovery (given that this is more dependent on the post-lesion integrity of the corticospinal tract and requires a higher degree of residual motor function after stroke to recover), gait abnormalities may often persist owing to the extent of gait pattern generators’ impairment (with particular regard to dorsiflexion strength) [43,44]. Therefore, several attempts to strengthen rehabilitation efficacy have been made using robotic devices and NIBS. In particular, RAR is thought to enhance brain plasticity mechanisms sustaining motor function recovery through intensive, repetitive, task-oriented, and assisted-as-needed motor practice, whereas NIBS mainly triggers a synaptic plasticity mechanisms potentiation via both an open loop and closed loop [43,44].

Concerning upper limb post-stroke rehabilitation, the available RCTs suggest that NIBS with RAR intervention was not superior to stand-alone RAR concerning the improvement in upper limb body structure and function [3,45]. However, this lack of clear evidence may depend on several methodological discrepancies among studies, including NIBS paradigms and motor training protocols. The NIBS protocols included in this review were mainly conducted according to an interhemispheric competition model, which posits that suppressing the excitability of the hemisphere not affected by stroke will enhance recovery by reducing interhemispheric inhibition of the stroke hemisphere, thus using bihemispheric stimulation, or to a vicariation model, which links functional recovery to the structural reserve spared by the lesion, thus targeting specific central nervous system areas, including the spinal cord and the cerebellum [43]. Despite this, there were large instrumental-related (stimulation number and duration, timing for RAR), patient-related (different levels of impairment) [42,46], and lesion location differences [47]. Furthermore, as is well known in the literature, intersubject and intrasubject variability consistently affect the potential of NIBS as a therapeutic tool [48,49,50]. Neuronavigated NIBS and electroencephalogram monitoring of NIBS effects in real time may be a viable option to decrease the variability in NIBS effects [48]. Contemporary, objective neuromarkers may be used to personalize NIBS paradigms [51].

A still unsolved issue regards the timing of NIBS to RAR. NIBS-RAR coupling is estimated to provide patients with a consistent amount of plasticity, which fosters recovery mechanisms. However, the order of application could have differential effects since the mechanisms of action of NIBS are different during and after stimulation [52]. Whether stand-alone NIBS and RAR can increase corticomotor excitability, the effects of these interventions are not necessarily additive; actually, subsequent practice-related synaptic activity might mask or even reverse NIBS aftereffects [53,54,55]. On the other hand, adding much more plasticity using NIBS during RAR may entrain a ceiling effect so that the impact of high-intensity motor practice on performance improvement may exceed the magnitude of the NIBS effects [56]. Furthermore, NIBS may act better on the retention of improved motor performance (as post-RAR NIBS sessions) or for preparing the brain for motor training (priming as pre-RAR NIBS intervention) than on motor training potentiation (as on-RAR NIBS session) [31,33,38]. However, the role of NIBS for shorter periods and the type of RAR best fitting with NIBS (including targeted joints and bilaterality of RAR intervention, especially for UL) remain unsolved [28,38,57]. Finally, the used outcome measures may be poorly sensitive to detect NIBS’s contribution to RAR [32,57,58,59].

Concerning lower limb studies, the available data suggest that the combined approach outperformed stand-alone RAR with regard to gait recovery. In particular, NIBS targeting the affected brain area for the lower limb or both the cerebellum and the spinal segment at the D10 level, in addition to RAR, improved walking ability (FAC) and capacity (6MWT). However, many unsolved issues remain, particularly regarding stimulation setup (intensity, dosage, session number, electrode positioning, even affected or contralesional hemisphere), type of stimulation, and timing concerning RAR delivery [33,38,60,61]. In this regard, we found that on-RAR and post-RAR bihemispheric tDCS improved patients’ balance and gait endurance compared to pre-RAR tDCS (contrarily to what was reported in a recent systematic review and consistently with previous reports) [62]. Further studies are nevertheless needed to shed light on the right timing of NIBS-RAR coupling.

Another significant issue to consider is the multiple localization of gait control mechanisms across the central nervous system, including the cerebellum and the spinal cord. There is some evidence that contemporary targeting of all of these structures might result in a functional improvement in patients with stroke through an increase in motor unit recruitment [63,64], a change in lower motor neuron responsiveness (in particular, they may become more responsive to synaptic activation but less prone to generate spontaneous activity that inhibits interneuronal networks) [63,64], a potentiation of cerebrum–cerebellum pathways that may be involved in a functional reorganization of motor networks following stroke and may substitute motor or cognitive systems in supratentorial stroke by playing as a non-lesioned entry [65,66].

### Strength and Weakness of NIBS-RAR Coupled Intervention

Consistent with the cardinal issue that neuroplasticity is the key process in motor function relearning, targeting specific brain areas with NIBS during RAR can further improve brain metabolism and neural–synaptic activity. In line with this principle, TMS and tDCS are aimed at stimulating an appropriate brain area by depolarizing neurons and activating excitatory action potentials, which inhibits/excites cortical neurons [67]. This principle is corroborated by the clinical practice that NIBS can magnify RAR aftereffects in post-stroke patients. The NIBS-added improvement likely depends on the capability of NIBS to focus on the brain plasticity strengthening induced by sustained motor practice using RAR, thus further fostering motor function recovery.

However, the exact mechanism by which TMS works is still partially unclear. We can consider three levels of action: molecular, cellular, and network. The levels’ functionality depends on several factors related to the individual neurobiology (including an individual’s excitability threshold) and the stimulation setup (including intensity, dose, and stimulation location), whose standardization is crucial across experiments [68]. In particular, magnetic pulses influence the ongoing activity of those neurons located horizontally in a surface parallel to the TMS coil [69]. The rapid change in the magnetic field induces circular electric currents; thus, the current flow is parallel to the coil and to the scalp on which the coil is placed flat, leading to axonal depolarization and the activation of cortical pathways, up to some subcortical structures, including the thalamus and the basal ganglia [50]. Motor cortex activation by TMS causes different descending volleys in the corticospinal tracts (including the earliest D-wave by direct stimulation of the neurons of the pyramidal pathway, and the I-wave by trans-synaptic stimulation of the pyramidal pathway) [70], whose motor unit recruitment follows the principle of size, from the smallest to the largest one [71]. Repeatedly stimulating the cortex leads to functional changes in synapses, mainly long-term potentiation and long-term depression, at both presynaptic and postsynaptic level [72]. These mechanisms, variably shaped, exert neurorestorative effects, leading to structural neuronal and network changes [17,73,74], with a relevant role of BDNF [75,76,77].

The neurobiological effects of tDCS are similarly partially known. tDCS consists of applying a low-intensity current (1–2 mA) between two or multiple small electrodes applied over the scalp [78]. The effects are mainly, but not only, influenced by the electrode polarity, with consequent modification of the resting membrane potential. Usually, anodal stimulation induces depolarization and increases cortical excitability, whereas cathodal stimulation produces hyperpolarization and decreases cortical excitability [79,80,81,82]. Both stimulations can be applied simultaneously on opposite targets, according to the interhemispheric inhibitory competition model [83,84,85], producing an interhemispheric rebalancing effect [86].

Consistent with these premises, coupled NIBS-RAR intervention may help in a post-stroke rehabilitation setting, although we have to acknowledge that all patients were also treated with conventional physiotherapy, which may have contributed to the recovery. Actually, conventional physiotherapy acts similar to RAR as a bottom-up approach, although robot-assisted repetition can improve gait performance and upper limb movement precision and reproducibility more than conventional physiotherapy. Notwithstanding this, it has been shown by Cochrane reviews [19,20] that RAR and conventional physiotherapy are not significantly different concerning daily life activities and arm functions, despite the greatest effects being appreciable within 3 months post-stroke. Concerning gait recovery, RAR increases the chance of independent walking (but not walking velocity and capacity) at the end of the treatment but not at the follow-up, regardless of the stroke stage, the pre-stroke status, and the type of the device employed.

One could argue that NIBS could also have positive effects when coupled with conventional therapy. Actually, several works assessed NIBS coupled with conventional physiotherapy as compared to stand-alone for either upper or lower limbs, showing the coupled intervention as an effective strategy to improve motor function recovery in post-stroke patients [2,87]. No studies directly compared RAR, NIBS, and conventional physiotherapy. However, it can be argued that RAR allows a better standardization of the rehabilitation exercises concerning, above all, the timing of execution. This is critical if we consider that NIBS stimuli work in the temporal path of milliseconds, thus being basilar regarding associative plasticity mechanisms, which are critical concerning synaptic plasticity strengthening and motor relearning; therefore, we may speculate that RAR is more suitable for NIBS compared to conventional physiotherapy concerning plasticity mechanisms’ triggering. This justifies the growing interest of the scientific community in the evaluation of the effects of RAR coupled with NIBS in stroke [88], and some preliminary, convincing data suggest a solid rationale for its implementation in advanced rehabilitation settings. NIBS can strengthen the deficitary brain network within the lesion site and inhibit the overactive brain networks neighboring the brain lesion. This NIBS-dependent bihemispheric effect was originally proven in experimental models employing intracortical microstimulation, achieving a rapid cortical reorganization of motor representation [89]. In addition, there is robust evidence that cortical stimulation can modulate cortical excitability and the motor responses evoked from the stimulated cortex, increase the dendritic density in the stimulated cortex, favor the reorganization of representational maps in the stimulated cortex, and lead to the synchronization and spreading of the perilesional neuronal activity supporting a major rewiring of far-to-distant connections, including transcallosal loops [16,21,90,91,92,93,94,95,96].

Although promising, conjugating NIBS with RAR, as well as the single implementation of such tools, requires device and instrument availability, personnel trained in the use of robots and NIBS, and time, space, and human resources [97]. In addition to these factors, a higher degree of patient compliance is mandatory to afford NIBS and/or RAR. Furthermore, no neurophysiological assessment with TMS was performed to assess cortical excitability and brain connectivity before and after treatments (with a few exceptions) [33,38]. Finally, the magnitude and duration of NIBS-RAR aftereffects depend on many variables related to instrumentation and the stimulation paradigm, and the setting and patient subjectivity to NIBS. Notwithstanding this, NIBS was safe in post-stroke settings [3,9,39,40].

## 5. Final Remarks and Conclusions

Post-stroke rehabilitation is focused on the relearning of the lost skills aimed at regaining independence, decreasing the disability burden, and improving the quality of life. Many novel strategies have been introduced in neurorehabilitation to facilitate the achievement of the abovementioned fundamental goals, including RAR and NIBS. The present scoping review provided a broad overview of studies coupling NIBS with RAR, illustrating the general pros and cons in rehabilitation practice that may influence future decisions in patient therapy and trigger future innovative clinical studies.

Despite the limited amount of scientific evidence, our review suggests that the combined approach has the potential to be beneficial to stroke patients more than the stand-alone treatments. This potential likely arises from the additional amount of plasticity offered by NIBS with regard to the reduction in the healthy brain hemisphere excitability, the reduction in the inhibition of the affected brain hemisphere, and the triggering of new, competitive, beneficial, inhibition balance patterns between the brain hemispheres. However, given there is still insufficient data in this field, mainly due to the overall limited sample size of the available RCTs (with heterogeneous properties of stroke concerning phase, stoke location, and extent) and a huge variability in approaches (both RAR and NIBS setup; follow-up duration), further research is needed to confirm the combined approach’s efficacy and translate it into clinical practice. In particular, more research is needed to maximize the effectiveness of existing protocols by optimizing stimulation dosage, intensity, and duration, by considering the brain state with EEG-triggered interventions, and by better characterizing the targeted stroke cohorts that may benefit. Furthermore, the timing of applying NIBS, limb targeting, parameters or types of NIBS, and subjects’ status (severity, phase of recovery—acute, subacute, or chronic) are other still unsolved issues. In order to suggest a routinely clinical application of combined NIBS-RAR [3,9,41], it will be necessary to conciliate the necessity of more homogenous rehabilitation protocols and therapeutic intervention designs tailored to every patient [44].

## Figures and Tables

**Figure 1 brainsci-12-01511-f001:**
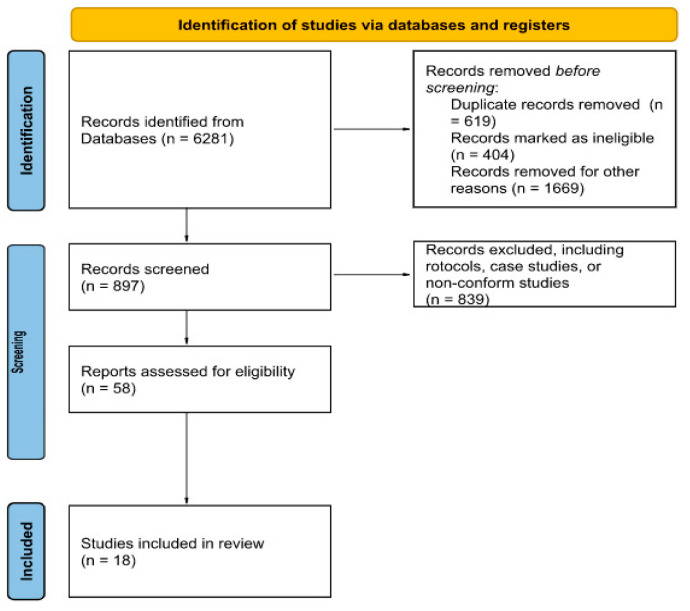
Study selection diagram flow.

**Table 1 brainsci-12-01511-t001:** Characteristics of upper limb studies.

Authors	Active-Control Group n	Device Type	Joint Task	NIBS	Outcomes ^#^
NIBS + RAR vs. RAR + sham NIBS comparison					
Ang 2015 [25]	10-9	MIT Manus (EE) ^1^	shoulder/elbow unimanual tasks	a/c-tDCS 10 sessions, 20-min before RAR for 2 weeks	active > sham in AL active = sham in BFS
Dehem 2018 [26]	11-10	REAplan robot (EE)	shoulder/elbow unimanual tasks	a/c-tDCS 1 session, 20-min during RAR (20-min) 1 day	**active > sham** in AL
Di Lazzaro 2016 [27]	8-9	InMotion2 * (EE) ^2^	shoulder/elbow unimanual tasks	continuous theta-burst stimulation 10 sessions on affected hemisphere before RAR for 2 weeks	**active > sham** in BFS
Edwards 2019 [28]	41-41	MIT Manus (EE) ^3^	entire arm unimanual tasks	a-tDCS 36 sessions, 20-min before RAR for 12 weeks	active = sham in BFS and AL
Hesse 2011 [29]	32-32	Bi-Manu Track (EE) ^4^	wrist/hand bimanual tasks	a-tDCS 30 sessions, 20-min at RAR beginning (400 movements) for 6 weeks	active = sham in BFS
	32-32	Bi-Manu Track (EE) ^5^	wrist/hand bimanual tasks	c-tDCS 30 sessions, 20-min at RAR beginning (400 movements) for 6 weeks	active = sham in BFS
Panker 2011 [30]	9-9	ReoGo(EE)	shoulder/elbow unimanual tasks	a-tDCS 22 sessions, 20-min at RAR beginning (60 min) for 2.5 weeks	**active > sham** in BFS **sham > active** in AL
Straudi 2016 [31]	12-11	ReoGo (EE)	shoulder/elbow unimanual tasks	a/c-tDCS 10 sessions, 30-min during the 30 min of RAR for 2 weeks	active = sham in BFS sham > active in AL
Tedesco Triccas 2015 [32]	12-11	Armeo^®^Spring (Ex)	whole arm unimanual tasks	a-tDCS 18 sessions, 25-min during the first 25 min of 75 min RAR for 8 weeks	**sham > active** in BFS sham > active in AL
Mazzoleni 2019 [34]	20-19	InMotion (EE)	wrist unimanual tasks	a-tDCS 30 sessions, 20-min during the treatment for 6 weeks	active = sham in BSF active > sham in AL
Timing of NIBS delivery during RAR					
Giacobbe 2013 [33]	12-12-12	InMotion3 (EE)	wrist unimanual tasks	a/c-tDCS 1 session, 20-min before vs. during vs. after training (with sham during the training) 1 day	after > before = during in movement speed before > during = after in movement smoothness after > before = during in speed reduction after = during = before in MEP increase
Different NIBSs’ comparison					
Ochi 2013 [35]	18-18	Bi-Manu Track (EE)	elbow/wrist bimanual tasks	a-tDCS vs. c-tDCS 5 sessions, 10-min during the treatment for 5 days	a-tDCS=c-tDCS in BFS and spasticity no effects on MAL

Legend: AL, activity limitation (assessed using Action Research Arm Test, Box and Blocks Test, or Wolf Motor Function Test); MI-BCI, motor imagery based brain–computer interface; BFS, body function and structure (assessed using Fugl-Meyer Assessment); * the commercial version of MIT Manus; MEP, motor evoked potential; MAL, motor activity log; EE, end effector; Ex, exoskeleton; a-tDCS, anodal transcranial direct current stimulation on affected M1 ref. contralateral orbit; c-tDCS, cathodal transcranial direct current stimulation on unaffected M1 ref. contralateral orbit. Adjunctive therapies: ^1^ 1-h RAR (using MI-BCI); ^2^ 960 RAR movements (active assistive); ^3^ 1 h RAR (1024 movement repetitions); ^4^ physical and occupational therapy; ^5^ physical and occupational therapy; ^#^ significant outcomes in bold characters.

**Table 2 brainsci-12-01511-t002:** Characteristics of lower limb studies.

Authors	Active-Control Group n	Device	NIBS	Outcomes ^#^
NIBS + RAR vs. RAR + sham NIBS comparison				
Danzl 2013 [36]	4-4	Lokomat (Ex)	c/a-tDCS (ref. supraorbit) 3 days/week sessions, 20-min before the training (20–40 min) for 4 weeks	active = sham in gait speed and balance
Geroin 2011 [37]	10-10-10	Gait Trainer 1 (Ex)	c/a-tDCS (ref. contralateral orbit) 5 days/week sessions, 7-min real vs. sham vs. no NIBS during the training (50-min) ^1^ for 2 weeks	active = sham **> no NIBS** in gait endurance active = sham **> no NIBS** in gait speed
Seo 2017 [41]	10-11	Walkbot_S (Ex)	c-tDCS (ref. contralateral orbit) 5 days/week sessions, 20-min during the training (45 min) for 2 weeks	**active > sham** in gait endurance
Timing of NIBS delivery during RAR				
Naro 2020 [38]	9-15-18	Lokomat^®^Pro(Ex)	c/a-tDCS (ref. contralateral CMA) 6 days/week sessions, 10-min before vs. during vs. after training (60-min) ^2^ for 8 weeks	during = after **> before** in gait endurance, fall risk, and gait performance during = after = before in gait speed, disability burden, and gait performance
Different NIBSs’ comparison				
Picelli 2015 [39]	10-10-10	G-EO System (EE)	a-tDCS (ref. contralateral orbit) + sham tsDCS sham tDCS + tsDCS* a-tDCS (ref. contralateral orbit) + tsDCS* 5 days/week sessions, 20-min during the training (20 min) for 2 weeks	**active > sham** in gait endurance up to two weeks but not up to one month after treatment
	10-10	G-EO System (EE)	tcDCS ** +tsDCS * a-tDCS (ref. ipsilateral orbit) + tsDCS * 5 days/week sessions, 20-min during the training (20 min) for 2 weeks	**active > sham** in gait endurance up to two weeks but not up to one month after treatment
Picelli 2019 [40]	20-20	G-EO System (EE)	tcDCS ** + tsDCS * tcDCS *** + tsDCS * 5 days/week sessions, 20-min during the training (20 min) for 2 weeks	active = sham in gait endurance
Leon 2017 [42]	9-17-23	Gait Trainer or Lokomat (Ex)	tDCSLEG tDCSHAND tDCSNO 5 days/week session, 20-min during the training (30–45 min) for 4 weeks	tDCS_LEG_ = tDCS_HAND_ = tDCS_NO_ in improving gait speed and gait performance

Legend: a-tDCS, anodal transcranial direct current stimulation on affected cortical motor areas (CMA) controlling lower limb; c-tDCS, cathodal transcranial direct current stimulation on unaffected cortical motor areas controlling lower limb; * tsDCS, cathodal trans-spinal direct current stimulation with D10 ref. ipsilesional shoulder; ** tcDCS, cathodal transcerebellar direct current stimulation on contralesional cerebellar hemisphere ref. contralesional buccinator muscle; *** tctDCS, cathodal transcerebellar direct current stimulation on ipsilesional cerebellar hemisphere ref. contralesional buccinator muscle; adjunctive therapies: ^1^ 30 min of lower limb muscle strengthening and joint mobilization, ^2^ physical rehabilitation program (1 h); tDCS_LEG_ a-tDCS (affected CMA-leg ref. contralateral orbit); tDCS_HAND_ a-tDCS (affected M1 ref. contralateral orbit); tDCS_NO_ no tDCS; ^#^ significant outcomes in bold characters.

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
