# Peer review of "Improving Upper Limb and Gait Rehabilitation Outcomes in Post-Stroke Patients: A Scoping Review on the Additional Effects of Non-Invasive Brain Stimulation When Combined with Robot-Aided Rehabilitation"

_brainsci, 2022, doi:10.3390/brainsci12111511_

Round 1

Reviewer 1 Report

This scoping review tried to identify the effects of the treatment combining NIB with RAR. However, because of the heterogeneity of the included studies and the organization of this scoping review, this scoping review may not make contribution to the fields of RAR and NIB. It would be difficult for readers to understand more about NIB and RAR from this scoping review.

According to Grant and Booth (2009), Scoping reviews are "preliminary assessment of potential size and scope of available research literature. Aims to identify nature and extent of research evidence (usually including ongoing research).” The present scoping review, however, did not provide enough details and information and did not organize the literature well. It did not present “nature and extent of research evidence”

Below are some comments and suggestions

Because the authors would like to discuss the effects of the combination of two treatments, NIB and RAR. They should clearly list the comparison group (sham group) and organize the papers based on the purposes of the studies. It seems that some studies tried to investigate the “additional” effects of NIB but others tried to compare “NIB and RAR” to no treatment.

The authors should also clarify the differences between RAR and conventional rehabilitation. The conventional rehabilitation should be able to provide “bottom-up effects” as RAR.

The authors should also provide more information about the treatment goals or mechanisms of RAR and NIB. This scoping review seems to hint most of the RAR aims to provide partial weight bearing for affected lower limb, so the patients can walk with a better gait pattern. But this scoping review did not provide information about why the RAR was used for upper extremity.

The authors should also discuss, not just list, the hypotheses/theories of the NIB application. For examples, what are their theories that they based on to apply NIB before, after or during RAR, and the theories of the locations of NIB application.

Author Response

This scoping review tried to identify the effects of the treatment combining NIB with RAR. However, because of the heterogeneity of the included studies and the organization of this scoping review, this scoping review may not make contribution to the fields of RAR and NIB. It would be difficult for readers to understand more about NIB and RAR from this scoping review. According to Grant and Booth (2009), Scoping reviews are "preliminary assessment of potential size and scope of available research literature. Aims to identify nature and extent of research evidence (usually including ongoing research).” The present scoping review, however, did not provide enough details and information and did not organize the literature well. It did not present “nature and extent of research evidence” Below are some comments and suggestions

We thank the reviewer for this interesting food for though. It however necessary to remember that scoping reviews are an ideal tool to determine the scope or coverage of a body of literature on a given topic, also providing the readers with a broad (in our case) overview of a given topic. In addition, scoping reviews are useful for paving the way to what of a given topic should be more developed in more precise systematic reviews. Scoping review has thus a more practical cut concerning the field and the way the research has been conducted. We therefore believe that our review presents the nature and extent of research evidence concerning a such specific topic, i.e., conjugating RAR and NIBS concerning post-stroke motor outcome improvement, as it focuses on identifying the types of available evidence in the field, clarifying key concepts/ definitions in the literature, examining how research is conducted on the topic, identifying key characteristics or factors related to the topic, and identifying and analyzing knowledge gaps. As we were not aimed at addressing the feasibility, appropriateness, meaningfulness or effectiveness of a certain treatment or practice (which would require a systematic review, but rather at identifying certain characteristics/concepts in papers or studies, and at mapping, reporting, and discussing the resulting characteristics/concepts, we planned and performed our scoping review in the present way. Besides, we want to thank the reviewer for his/her constructive criticisms and the useful suggestions to improve the quality of our review. We addressed as follows the reviewer’s concerns:

  • Because the authors would like to discuss the effects of the combination of two treatments, NIB and RAR. They should clearly list the comparison group (sham group) and organize the papers based on the purposes of the studies. It seems that some studies tried to investigate the “additional” effects of NIB but others tried to compare “NIB and RAR” to no treatment.

All the studies included in the present review were aimed at investigating the effects of NIBS+RAR concerning upper limb motor function and gait as the main outcome compared to a RAR control group. We now better specified that patients were provided with RAR associated with NIBS, including a control group with characteristics comparable to the experimental group, which was provided with RAR paired with sham NIBS. This design was consistent with the purpose of the included studies of assessing whether NIBS and RAR was superior to RAR alone. However, one study [35] included also a third control group (no NIBS), whereas some other studies also included a comparative analysis between different types [33,37,38,40] and timing of delivery of NIBS [31,36]. All included studies provide all the enrolled groups also with conventional therapy (including muscle strengthening, joint mobilization exercises, and a comprehensive physical rehabilitation program). Result reporting was revised accordingly.

  • The authors should also clarify the differences between RAR and conventional rehabilitation. The conventional rehabilitation should be able to provide “bottom-up effects” as RAR.
  • The authors should also provide more information about the treatment goals or mechanisms of RAR and NIB. This scoping review seems to hint most of the RAR aims to provide partial weight bearing for affected lower limb, so the patients can walk with a better gait pattern. But this scoping review did not provide information about why the RAR was used for upper extremity.

Affirming that “This scoping review seems to hint most of the RAR aims to provide partial weight bearing for affected lower limb, so the patients can walk with a better gait pattern” is rather reductive. RAR, including exoskeletons and end-effector devices, provides patients with intensive, repetitive, assisted-as-needed, and task-oriented motor practice, which are critical to boost neural plasticity mechanisms sustaining motor function recovery. This concept is largely supported by the current literature. The same though is extensible to upper limb rehabilitation. The relevant information concerning treatment goals or mechanisms of RAR and NIBS are already available in the introduction and we further expanded these, as suggested by the reviewer.

  • The authors should also discuss, not just list, the hypotheses/theories of the NIB application. For examples, what are their theories that they based on to apply NIB before, after or during RAR, and the theories of the locations of NIB application.

Accordingly, we added more details on this relevant issue.

Kindest regards,

The authors

Reviewer 2 Report

The authors did a scoping review including 18 studies on combined effects of NIBS and RAR in stroke patient rehabilitation (separated for upper and lower limb studies). They conclude that tDCS and RAR in combination are promising but not yet largely recommendable as a systematic approach for stroke rehabilitation, as there is not enough data about it.

Major:

My first concern deals with the term neuroplasticity, already mentioned in the title of this scoping review. Did the studies really study neuroplasticity, or is it rather motor function? According for example Puderbaugh and Prabhu neuroplasticity involves adaptive structural and functional changes to the brain. They define it as “the ability of the nervous system to change its activity in response to intrinsic or extrinsic stimuli by reorganizing its structure, functions, or connections.” I don’t think that the included studies examined neuroplasticity per se. To measure neuroplasticity for example MRI techniques will be necessary or at least TMS.

Second main concern is the description of NIBS. I suggest to better specify the difference between a-tDCS, c-tDCS at different locations and TMS and the different impact (and possible limitation) on stroke rehabilitation.

Was the scoping review performed according to the Preferred Reporting Items for Scoping Reviews (PRISMA-ScR) guideline? If not, why not?

Minor:

Introduction

-          Please ad ‘both “of” these’  or delete “these” in the sentence: re-learning [14]. Both these approaches act on the spontaneous…

-          Sentence “Specifically, these interventions are postulated to: (i) favor remote structures' re-connection to the site of injury following the diaschisis period (i.e., a temporary period of depressed metabolism and blood flow), including peri-lesional cortex, spared areas in the injured hemisphere,…” Which interventions are meant: bottom-up approaches? Please use the same wording for the intervention/approaches throughout the hole manuscript.

Material and Methods:

-          Please provide the exact initial search string of at least one databank

-          Sentence “…(ii) improvement in upper limb motor function and gait as the main outcome; and (iii) use of RAR and nNIBS in combination.” What is nNIBS? Typo?

-          Why were cerebellar strokes not included?

-          Sentence “tDCS setup varied among studies for electrode dimension (12.56 cm2 for cerebellar tDCS, 23.75 cm2 for spinal tDCS, 25 vs. 35 cm2 for tDCS),…” 25 vs 35 cm2 for tDCS over M1?

Results

-          Table 1 in the NIBS column: 1. stimulation location. Stimulated every study over M1? 2. I don't understand the numbers before the stimulation duration, please explain.

-          Table 2 in the outcome column: was it gait or balance in the Tinetti? Perhaps it would be helpful in Table 1 and 2 to mention the outcome domain instead of the measured test.

-          Please mark the studies in the Tables with significant outcome results.

-         Subjects were evaluated for body function and structure per Fugl Meyer Assessment but one [24], the activity limitation as per Box and Blocks Test, Wolf Motor Function Test, and Action Research Arm Test but three [23,25,27], and the mobility index.à I don’t understand this sentence. Please reformulate.

Discussion

-          Please remove the parentheses before and after NIBS before Ref [43-44].

-          What does RAGT and ICMS mean?

Author Response

The authors did a scoping review including 18 studies on combined effects of NIBS and RAR in stroke patient rehabilitation (separated for upper and lower limb studies). They conclude that tDCS and RAR in combination are promising but not yet largely recommendable as a systematic approach for stroke rehabilitation, as there is not enough data about it.

 Major:

  • My first concern deals with the term neuroplasticity, already mentioned in the title of this scoping review. Did the studies really study neuroplasticity, or is it rather motor function? According for example Puderbaugh and Prabhu neuroplasticity involves adaptive structural and functional changes to the brain. They define it as “the ability of the nervous system to change its activity in response to intrinsic or extrinsic stimuli by reorganizing its structure, functions, or connections.” I don’t think that the included studies examined neuroplasticity per se. To measure neuroplasticity for example MRI techniques will be necessary or at least TMS.

We thank the reviewer for this food for thought, on which we agree. Consistently we modified the title of the paper to make it more consistent with the aim of our scoping review, i.e., to assess the usefulness of NIBS and RAR in combination to improve upper limb and gait related outcomes in post-stroke patients

  • Second main concern is the description of NIBS. I suggest to better specify the difference between a-tDCS, c-tDCS at different locations and TMS and the different impact (and possible limitation) on stroke rehabilitation.

Accordingly, we added more information of these issues.

  • Was the scoping review performed according to the Preferred Reporting Items for Scoping Reviews (PRISMA-ScR) guideline? If not, why not?

We followed the PRISMA-ScR guidline, as now better specified in the text. We also included the PRISMA-ScR) Checklist as supplementary file

Minor:

Introduction

  • Please ad ‘both “of” these’  or delete “these” in the sentence: re-learning [14]. Both these approaches act on the spontaneous…

Corrected

  • Sentence “Specifically, these interventions are postulated to: (i) favor remote structures' re-connection to the site of injury following the diaschisis period (i.e., a temporary period of depressed metabolism and blood flow), including peri-lesional cortex, spared areas in the injured hemisphere,…” Which interventions are meant: bottom-up approaches? Please use the same wording for the intervention/approaches throughout the hole manuscript.

The sentence was entirely revised since misleading.

Material and Methods:

  • Please provide the exact initial search string of at least one databank

Done.

  • Sentence “…(ii) improvement in upper limb motor function and gait as the main outcome; and (iii) use of RAR and nNIBS in combination.” What is nNIBS? Typo?

Corrected into NIBS

  • Why were cerebellar strokes not included?

We focused on supratentorial stroke to be consistent with previous literature revision.

  • Sentence “tDCS setup varied among studies for electrode dimension (12.56 cm2 for cerebellar tDCS, 23.75 cm2 for spinal tDCS, 25 vs. 35 cm2 for tDCS),…” 25 vs 35 cm2 for tDCS over M1?

Corrected.

Results

  • Table 1 in the NIBS column:
    • stimulation location. Stimulated every study over M1?

The studies idluded in the review targeted the affected or unaffected M1 ref. contralateral orbit or the cortical motor areas (CMA) controlling lower limb (ref. to different references)

  • I don't understand the numbers before the stimulation duration, please explain.

We were referring to the number of sessions and their duration, as now better explicated.

  • Table 2 in the outcome column: was it gait or balance in the Tinetti? Perhaps it would be helpful in Table 1 and 2 to mention the outcome domain instead of the measured test.

Amended as suggested

  • Please mark the studies in the Tables with significant outcome results.

Done.

  • “Subjects were evaluated for body function and structure per Fugl Meyer Assessment but one [24], the activity limitation as per Box and Blocks Test, Wolf Motor Function Test, and Action Research Arm Test but three [23,25,27], and the mobility index.” I don’t understand this sentence. Please reformulate.

Checked and corrected.

 Discussion

  • Please remove the parentheses before and after NIBS before Ref [43-44].

Done.

  • What does RAGT and ICMS mean?

Corrected into RAR and, respectively, intracortical microstimulation.

Kindest regards,

The authors

Reviewer 3 Report

- The two primary therapies for post-stroke rehabilitation are robot-aided rehabilitation (RAR) and non-invasive brain stimulation (NIBS). The effectiveness of the two strategies when used together has not yet been thoroughly proven. Combining these therapies, which both increase brain plasticity to encourage recovery, is crucial because it increases the possibility for rehabilitation to minimize the limitations in daily living activities and the quality of life after a stroke. This evaluation sought to assess the effectiveness of NIBS combined with RAR in enhancing rehabilitation results for adult stroke patients with upper-limb and gait motor dysfunction. In this review the authors incorporated eighteen clinical studies. Regarding the technical aspects of robotic devices and NIBS protocols, all investigations were very diverse. The studies did note a general improvement in body composition and function as well as activity limitation for the upper limb, but these differences between the active and control groups were not statistically significant. The active group outperformed the control group in terms of gait training procedures, enhancing walking capacity and recovery. This review suggests that NIBS and RAR together are promising but not yet generally recommended as a systematic approach for stroke rehabilitation due to a lack of data.

the authors squeezed the work well however I have the following reservations:

- In the search strategy whether the keywords used simultaneously or using the AND function of WoS? The authors need to systematically write the keywords.

- Reports means what in the search/discard category in the flow diagram?

- Which year were selected while searching the studies?

- It is not necessary that anodal tDCS will always result positive, how do the authors interpret the negative or nill effects of modulation? It was observed that sometimes cathodal tDCS resultantly better. 

- Conclusions are not enough to justify the short review. Please elaborate more. 

- The flow of the paper seems like a research article not the review article. It is not correct. The authors needs to see the published ones from to get an idea. for example:   Non-invasive transcranial electrical brain stimulation guided by functional near-infrared spectroscopy for targeted neuromodulation: A review.  

-What is the new findings/suggestions that authors can interpret from the review? 

Without solving the above reservation, the manuscript is not ready for publication. 

- The granular findings of the review papers are not seen throughout the manuscript. How the authors interpret the findings of the study? 

Author Response

The two primary therapies for post-stroke rehabilitation are robot-aided rehabilitation (RAR) and non-invasive brain stimulation (NIBS). The effectiveness of the two strategies when used together has not yet been thoroughly proven. Combining these therapies, which both increase brain plasticity to encourage recovery, is crucial because it increases the possibility for rehabilitation to minimize the limitations in daily living activities and the quality of life after a stroke. This evaluation sought to assess the effectiveness of NIBS combined with RAR in enhancing rehabilitation results for adult stroke patients with upper-limb and gait motor dysfunction. In this review the authors incorporated eighteen clinical studies. Regarding the technical aspects of robotic devices and NIBS protocols, all investigations were very diverse. The studies did note a general improvement in body composition and function as well as activity limitation for the upper limb, but these differences between the active and control groups were not statistically significant. The active group outperformed the control group in terms of gait training procedures, enhancing walking capacity and recovery. This review suggests that NIBS and RAR together are promising but not yet generally recommended as a systematic approach for stroke rehabilitation due to a lack of data. The authors squeezed the work well however I have the following reservations:

  • In the search strategy whether the keywords used simultaneously or using the AND function of WoS? The authors need to systematically write the keywords.

Amended.

  • Reports means what in the search/discard category in the flow diagram?

Flow diagram reporting was revised since misleading

  • Which year were selected while searching the studies?

No publication date restriction was imposed. However, the included studies were published between 2011 and 2020.

  • It is not necessary that anodal tDCS will always result positive, how do the authors interpret the negative or nill effects of modulation? It was observed that sometimes cathodal tDCS resultantly better. 

Amended, as correctly pointed out by the reviewer.

  • Conclusions are not enough to justify the short review. Please elaborate more.

We further expanded this sections, as suggested by the reviewer. 

  • The flow of the paper seems like a research article not the review article. It is not correct. The authors needs to see the published ones from to get an idea. for example:   Non-invasive transcranial electrical brain stimulation guided by functional near-infrared spectroscopy for targeted neuromodulation: A review.

We thank the reviewer for this suggestion. However, the review was written consistently with the PRISMA-ScR guideline and previous systematic reviews published on the topic (Bressi et al. 2022; Reis et al. 2021)

  • What is the new findings/suggestions that authors can interpret from the review? Without solving the above reservation, the manuscript is not ready for publication.

We further expanded the conclusion section, as suggested by the reviewer. 

  • The granular findings of the review papers are not seen throughout the manuscript. How the authors interpret the findings of the study?

It was not necessary to add granular findings of the reviewed papers as this was not a systematic review. We believe that the summary of the main outcome measures reported in table 1 and 2 is sufficient for a scoping review to provide the readers with a broad overview of the topic. This will be carried out in a subsequent systematic review that we are planning.

Kindest regards,

The authors

Round 2

Reviewer 1 Report

This revised manuscript has provided some missing information but is still not organized. Below are some suggestions.

 1.      Since most of the papers they reviewed are “RAR+NIBS” vs RAR. They actually discussion the additional effects of NIBS, when combined with RAR. They should clarity this on the title and in the introduction.

2.     In the discussion, they stated that the RAR and conventional physiotherapy shared the same recovery mechanisms. Then, what is the difference between conventional PT and RAR? They should clarify this in the introduction

l   In addition to repeatedly stating that RAR is an advanced rehab protocol (intensive, repetitive, assisted-as-needed and task-oriented motor practice), they should provide the information about what is the uniqueness of RAR (conventional PT could be intensive, repetitive, assisted-as-needed and task-oriented motor practice)

l   They should clarify that whether RAR has better effects than conventional physiotherapy. They only stated that “Actually, robot-assisted repetition with electromechanical trainer can improve gait performance and upper limb movement precision and reproducibility”, But is RAR better than conventional PT?

l   Why do the studies want to combine NIBS with RAR but not with conventional PT? Did they believed NIBS would have better effects with RAR?

3.      There were some conflicting statement related to the results of the effects of NIBS when combined with RAR, which is their main research questions. They should re-organize the results and discussion.

l   In the last paragraph of results, it seems no difference between active and control. They said “Overall, the studies reported a global improvement in body structure and function and activity limitation for the upper limb, which were, however, non-significant between the active and control group, with some exceptions

l   In the discussion, it seems the effects are different between upper extremity and lower extremity.

Concerning upper limb post-stroke rehabilitation, the available RCTs do not sufficiently suggest that NIBS adds significantly to RAR

Concerning gait recovery, the available data suggest that NIBS targeting the affected brain area for the lower limb or both the cerebellum and the spinal segment at the D10 level, in addition to RAR, improves walking ability (FAC) and capacity (6MWT).

However, in the last paragraph of discussion, they stated “Overall, such approaches induce a plasticity increase associated with an increase in motor recovery by either facilitating the ipsilesional hemisphere or inhibiting the contralesional one, depending on the stroke phase.” But they even did not discuss how the stroke phases affect the effects, and how they get the conclustion. They did not provide neurophysiological results in the result section.

In the conclusion, they said “Despite the limited amount of scientific evidence, our review shows that the combined approach is mostly beneficial to patients as compared to stand-alone treatments.”

It is confusing that whether NIBS has significant effects.

4.      As a scoping review, this manuscript should not only describe the differences of the protocols in the papers or the possible mechanisms or only indicate problems of the previous papers. They should provide some suggestions that what kind of study design may have better effects of NIBS+RAR, such as the timing of applying NIBS, upper limb or lower limb, parameters or types of NIBS, subjects status (severity, phase of recovery (acute, subacute or chronic)

Author Response

This revised manuscript has provided some missing information but is still not organized. Below are some suggestions.

We want to thank the reviewer for the appreciation to our ms and the further useful suggestion to improve its quality.

Since most of the papers they reviewed are “RAR+NIBS” vs RAR. They actually discussion the additional effects of NIBS, when combined with RAR. They should clarity this on the title and in the introduction.

We agree with the reviewer's concern. The title and the introduction were amended accordingly.

In the discussion, they stated that the RAR and conventional physiotherapy shared the same recovery mechanisms. Then, what is the difference between conventional PT and RAR? They should clarify this in the introduction. In addition to repeatedly stating that RAR is an advanced rehab protocol (intensive, repetitive, assisted-as-needed and task-oriented motor practice), they should provide the information about what is the uniqueness of RAR (conventional PT could be intensive, repetitive, assisted-as-needed and task-oriented motor practice)

We agree with the reviewer's point of view. We provided the missing information.

  • They should clarify that whether RAR has better effects than conventional physiotherapy. They only stated that “Actually, robot-assisted repetition with electromechanical trainer can improve gait performance and upper limb movement precision and reproducibility”, But is RAR better than conventional PT?

we revised this sentence since misleading. Actually, comparing RAR and conventional physiotherapy was not within the scope of this review

  • Why do the studies want to combine NIBS with RAR but not with conventional PT? Did they believed NIBS would have better effects with RAR?

We thank the reviewer for this interesting food for though. One could indeed argue that NIBS could also have positive effects when coupled with conventional therapy. Actually, several works assessed NIBS coupled with conventional physiotherapy as compared to stand-alone for either upper or lower limbs, showing the coupled intervention as an effective strategy to improve motor function recovery in post-stroke patients (Veldema, J., Gharabaghi, A. Non-invasive brain stimulation for improving gait, balance, and lower limbs motor function in stroke. J NeuroEngineering Rehabil 19, 84 (2022).         Cha, T. H., & Hwang, H. S. (2022). Rehabilitation Interventions Combined with Noninvasive Brain Stimulation on Upper Limb Motor Function in Stroke Patients. Brain sciences, 12(8), 994.). No studies directly compared RAR, NIBS, and conventional physiotherapy. However, it can be argued that RAR allows a better standardization of the rehabilitation exercises concerning, above all, timing of execution. This is critical if we consider that NIBS stimuli work in the temporal path os milliseconds, thus being basilar regarding associative plasticity mechanisms, which are critical concerning synaptic plasticity strengthening and motor relearning, therefore, we may speculate that RAR is more suitable for NIBS compared to conventional physiotherapy concerning plasticity mechanisms triggering.

There were some conflicting statement related to the results of the effects of NIBS when combined with RAR, which is their main research questions. They should re-organize the results and discussion.

Checked and corrected as per reviewer’s suggestion.

  • In the last paragraph of results, it seems no difference between active and control. They said “Overall, the studies reported a global improvement in body structure and function and activity limitation for the upper limb, which were, however, non-significant between the active and control group, with some exceptions

The sentence was revised because it was misleading. We better stated that “Overall, the studies reported that NIBS with RAR intervention was not superior to stand-alone RAR (with some exceptions that documented the superiority of the combined approach) concerning the improvement in upper limb body structure and function (Tables 1 and 2)[3,9]. On the contrary, the combine approach outperformed stand-alone RAR with regard to gait recovery”

  • In the discussion, it seems the effects are different between upper extremity and lower extremity. “Concerning upper limb post-stroke rehabilitation, the available RCTs do not sufficiently suggest that NIBS adds significantly to RAR” “Concerning gait recovery, the available data suggest that NIBS targeting the affected brain area for the lower limb or both the cerebellum and the spinal segment at the D10 level, in addition to RAR, improves walking ability (FAC) and capacity (6MWT).

The reviewer is right. We now better specified that NIBS with RAR intervention was not superior to stand-alone RAR concerning the improvement in upper limb body structure and function. On the contrary, the combined approach outperformed stand-alone RAR with regard to gait recovery

  • However, in the last paragraph of discussion, they stated “Overall, such approaches induce a plasticity increase associated with an increase in motor recovery by either facilitating the ipsilesional hemisphere or inhibiting the contralesional one, depending on the stroke phase.” But they even did not discuss how the stroke phases affect the effects, and how they get the conclustion. They did not provide neurophysiological results in the result section.

We agree with the reviewer’s point of view. We cannot draw this conclusion based on our review data. The sentence was deleted.

In the conclusion, they said “Despite the limited amount of scientific evidence, our review shows that the combined approach is mostly beneficial to patients as compared to stand-alone treatments.” It is confusing that whether NIBS has significant effects.

The sentence was revised because it was misleading.

As a scoping review, this manuscript should not only describe the differences of the protocols in the papers or the possible mechanisms or only indicate problems of the previous papers. They should provide some suggestions that what kind of study design may have better effects of NIBS+RAR, such as the timing of applying NIBS, upper limb or lower limb, parameters or types of NIBS, subjects status (severity, phase of recovery (acute, subacute or chronic)

We thank the reviewer for this suggestion that was adopted in the final remarks and conclusions paragraph.

Kindest regards,

the authors

Reviewer 2 Report

I am sorry. I had some problems with my e-mail account. Paper is ready to go. Thank you.

Reviewer 3 Report

it can be accepted in the current form

Author Response

it can be accepted in the current form

We thank the reviewer for the appreciation to our ms.

Kindest regards,

the authors